# In Vivo Absorption of Iron Complexes of Chondroitin Sulfates with Different Molecular Weights and Their Anti-Inflammation and Metabolism Regulation Effects on LPS-Induced Macrophages

**DOI:** 10.3390/foods14193356

**Published:** 2025-09-27

**Authors:** Qianqian Du, Jiachen Zheng, Fanhua Kong, Xiuli Wu, Chunqing Ai, Shuang Song

**Affiliations:** 1SKL of Marine Food Processing & Safety Control, National Engineering Research Center of Seafood, Collaborative Innovation Center of Seafood Deep Processing, National & Local Joint Engineering Laboratory for Marine Bioactive Polysaccharide Development and Application, Liaoning Key Laboratory of Food Nutrition and Health, School of Food Science and Technology, Dalian Polytechnic University, Dalian 116034, China; qianqian0506819@163.com (Q.D.); zhengjiachen0422@126.com (J.Z.); kkong0930@163.com (F.K.); acqdongying@163.com (C.A.); 2College of Pharmacy, Ningxia Medical University, Yinchuan 750004, China; wu.xiuli2005@163.com

**Keywords:** chondroitin sulfate, iron complex, in vivo absorption, tissue distribution, anti-inflammation, metabolic analysis

## Abstract

The present study investigated the effects of hierarchical molecular weights and iron chelation on the in vivo absorption and the inflammatory bioactivity of chondroitin sulfate (CS). Firstly, CS, chondroitin sulfate-iron complex (CS-Fe), and low-molecular-weight chondroitin sulfate-iron complex (LCS-Fe) were fluorescently labeled and characterized. Then, the plasma concentration–time profiles and fluorescence imaging results demonstrated that LCS-Fe was more efficiently absorbed into the bloodstream and showed a higher Cmax (415.16 ± 109.50 μg/mL) than CS-Fe (376.60 ± 214.10 μg/mL) and CS (135.27 ± 236.82 μg/mL), and it clearly accumulated in the liver. Furthermore, the anti-inflammatory effect of CS-Fe and LCS-Fe was assayed in LPS-induced macrophages, and LCS-Fe and CS-Fe both showed a better inhibitory effect on NO production, COX-2 and IL-1β gene expression levels compared to CS. Additionally, targeted metabolic analysis of macrophages using LC-MS/MS revealed that CS, CS-Fe, and LCS-Fe could reverse approximately one quarter of the LPS-induced differential metabolites, and the biosynthesis of valine, leucine, and isoleucine was the most significantly involved metabolic pathway. Notably, the molecular weight reduction and iron chelation could both enhance the bioavailability and anti-inflammatory efficacy of CS.

## 1. Introduction

Chondroitin sulfate (CS) is an anionic sulfated glycosaminoglycan (GAG) which is widespread in the extracellular matrix of animal connective tissues, such as cartilages, skin, blood vessels, tendons, and ligaments [1]. CS chain consists of repeating disaccharide units of →4)-β-GlcA-(1→3)-β-GalNAc-(1→, and its molecular weight (Mw) is commonly between 10 and 100 kDa. The structural diversity of CS primarily arises from variations in sulfation patterns and can be classified into five common types, including CS-O, CS-A, CS-C, CS-D, and CS-E [2], which contribute to its bioactivity and functional specificity. It has been reported that CS has various physiological and pharmacological activities, such as antioxidant, anti-inflammatory, anti-tumor, neuroprotective, and immunoregulatory activities [3,4,5,6]. Among them, the anti-inflammatory activity of CS has attracted great attention, and CS has already been used to treat some inflammation-related diseases, such as arthritis, gastric ulcer and colitis [2,7,8] through suppressing the activation of signaling pathways and down-regulating the inflammatory factors [5].

As the component of dietary supplements and functional foods, CS is typically administered orally; however, its structural characteristics, such as considerable Mw, charge density, hydrophilic groups, and poor fat solubility, lead to incomplete intestinal permeability, low bioavailability (0–13%), and a limited efficacy range [9]. The same situation has also been found for many bioactive polysaccharides, which are hardly degraded in the stomach, and their absorption primarily depends on the paracellular pathway, transcellular pathways, and carrier protein-mediated transport across intestinal epithelial cells [10]. Therefore, the uptake of these polysaccharides was determined by factors such as charge, Mw, dose, spatial structure, and physiological state.

Metal ions exhibit unique coordination capabilities with polysaccharides, potentially influencing their molecular structure, chain conformational transitions, and molecular size. Iron ions could serve as metal centers to form stable complexes with organic ligands, and compared to Al (III), Cu (II), and Zn (II) ions, Fe (III) exhibits a higher affinity for coordination groups and demonstrates stronger complexation capacity [11]. Moreover, Zhu et al. [12] reported that endogenous metal ions contribute to maintaining the triple helix structure of lentinan polysaccharides. In addition, the Fe (III) chelates with corn silk polysaccharides exhibited enhanced antioxidant activity and inhibition effects on α-glucosidase [13]. Furthermore, numerous studies have reported that combining iron ions with biocompatible materials, such as polysaccharides, polymers, dendrimers, and carbohydrates, reduces or avoids the systemic toxicity risks (cytotoxic, genotoxic, and hemolytic effects) of inorganic iron and better meets safety requirements as an oral biological agent [14,15]. In our previous study, iron ion complexes with CS and its low-molecular-weight derivatives have been prepared, and their molecular sizes have been greatly reduced due to the chelation of iron, which results in the enhanced transport efficiency in everted gut sac model.

However, in vivo absorption of chondroitin sulfate-iron complexes has not been evaluated, and whether the complexes still possess anti-inflammation properties as the original CS is not clarified. Therefore, in the present study, iron complexes of high-molecular-weight and low-molecular-weight chondroitin sulfate (CS-Fe and LCS-Fe) were prepared, and their absorption and distribution profiles in vivo were compared with those of unmodified CS. Furthermore, the anti-inflammatory effects of CS and its two iron complexes were evaluated using the RAW 264.7 macrophage cell model, and the underlying anti-inflammatory mechanism was analyzed through metabolomics.

## 2. Materials and Methods

### 2.1. Materials and Chemicals

Chondroitin sulfate (CAS: 9007-28-7; purity ≥ 95.0%) from Wuhan Dongkangyuan Technology Co., Ltd. (Wuhan, China) was constructed with repeating units of →3)-β-GalNAc (1→4)-β-GlcA (1→ with sulfate groups substituted at C-4 or C-6 of GalNAc, and its Mw and sulfate content were 68.60 kDa and 19.61 ± 0.55%, respectively [16]. FeSO_4_·7H_2_O was purchased from Tianjin Damao Chemical Reagent Co., Ltd. (Tianjin, China). TiO_2_ (P25; CAS: 13463-67-7) was purchased from Evonik Degussa Co., Ltd. (Shanghai, China). 30% H_2_O_2_ was purchased from Foshan Xilong Chemical Co., Ltd. (Guangzhou, China). Ammonium acetate (LC-MS grade) was purchased from Macklin Biochemical Co., Ltd. (Shanghai, China). 5-Aminofluorescein was supplied by Aladdin Co., Ltd. (Shanghai, China). Lipopolysaccharide (LPS) was obtained from Solarbio Biotechnology Co., Ltd. (Beijing, China). RAW 264.7 cell complete medium was purchased from Pricella Life Technology Co., Ltd. (Wuhan, China). Methanol (LC-MS grade) and acetonitrile (LC-MS grade) were provided by Sigma-Aldrich (Shanghai, China). The standard maintenance diet was purchased from Liaoning Changsheng Biotechnology Co., Ltd. (Benxi, China).

### 2.2. Preparations of CS-Fe and LCS-Fe

Chondroitin sulfate was dissolved in deionized water (5 mg/mL) and mixed with 20 mM FeSO_4_ with magnetic stirring. After stirring for 10 min, 20 mM H_2_O_2_ was added to the solution. The mixture was then centrifuged (10,000 rpm/min, 15 min), and the supernatant was dialyzed using a 300 Da cut-off membrane for 4 days. Finally, freeze-drying was performed to obtain the dry chondroitin sulfate-iron complex (CS-Fe).

According to our previous study [16], low-molecular-weight chondroitin sulfate-iron complex (LCS-Fe) was prepared using a photocatalysis-Fenton degradation method, with a reaction time of 60 min.

### 2.3. Fluorescent Labeling of CS and Its Iron Complexes

CS and its iron complexes (CS-Fe and LCS-Fe) were labeled with 5-aminofluorescein (5-AF) using a previously described method [17], with slight modifications. Specifically, CS, CS-Fe, and LCS-Fe (each 250 mg) were dissolved in deionized water (50 mL), subsequently oxidized with 160 μL of NaClO and 10 mL of a 2,2,6,6-tetramethylpiperidinyl-1-oxy (TEMPO) solution, and then freeze-dried. The oxidized samples (40 mg) and 5-AF (3 mg) were dissolved in the carbonate buffer (pH 9.5) and magnetically stirred at room temperature in dark for 48 h. Finally, the unlabeled 5-AF was removed by repeated dialysis with deionized water in dark. The labeled CS, CS-Fe, and LCS-Fe were freeze-dried and named F-CS, F-CS-Fe, and F-LCS-Fe, respectively.

### 2.4. Fluorescence Spectroscopy

The fluorescently labeled samples (F-CS, F-CS-Fe, and F-LCS-Fe) were dissolved in deionized water at a concentration of 1 mg/mL. The fluorescence intensities of the samples were measured using a fluorescence spectrophotometer (HitachiF-2700, Hitachi, Tokyo, Japan) at the excitation wavelength of 490 nm. The emission spectra were recorded over a wavelength range of 505 nm-600 nm. The slit width was 5 nm, the photomultiplier tube voltage was 400 V, and the scanning rate was 1500 nm/min.

### 2.5. Determination of Fluorescence Substitution Degree

The degree of fluorescent substitution was determined according to a reported method [18]. Briefly, 5-AF was dissolved in PBS (0.01 M, pH 7.2–7.4) at concentrations ranging from 1 μg/mL to 6 μg/mL. The fluorescence intensities at different concentrations were determined using a fluorescence microplate reader (Infinite F200 PRO, Tecan, Switzerland) at an excitation wavelength of 490 nm and an emission wavelength of 520 nm. The standard curve was then established concerning the fluorescence intensity plotted against the concentration of 5-AF. Subsequently, F-CS, F-CS-Fe, and F-LCS-Fe powders were dissolved in PBS at a concentration of 1 mg/mL. The fluorescence substitution degrees of F-CS, F-CS-Fe, and F-LCS-Fe were calculated referring to the standard curve.

### 2.6. High Performance Gel Permeation Chromatography (HPGPC)

The molecular size distribution was determined using high performance gel permeation chromatography (Waters Alliance 2695, Waters, Milford, MA, USA) equipped with a TSK-G5000PWXL chromatographic column, and coupled to a refractive index detector (Waters 2414, USA) for on-line detection. All samples were dissolved in deionized water at a concentration of 3 mg/mL. The mobile phase was ammonium acetate solution (20 mM). The injection volume, flow rate, and column temperature were 10 μL, 0.4 mL/min, and 30 °C, respectively. The dextran samples with Mw of 5, 25, 50, 410, and 670 kDa were applied as standards. The retention times of the chromatographic peaks were assigned to the abscissa, and the −lg Mw values were assigned to the ordinate to obtain the linear regression equation (y = 0.2877x − 11.017, *R*^2^ = 0.9985).

### 2.7. In Vivo Absorption and Tissue Distribution Assays

Specific pathogen-free (SPF) Kunming mice (male, 28–30 g) were purchased from Liaoning Changsheng Biotechnology Co., Ltd. (Benxi, China). Animal experiments complied with the ARRIVE (Animal Research: Reporting of In Vivo Experiments) guidelines and were carried out in accordance with the National Research Council’s Guide for the Care and Use of Laboratory Animals, and the experiment protocol was approved by Animal Ethics Committee of Dalian Polytechnic University (DLPU2024DT009). Briefly, all animals were housed in an SPF-graded facility with 12/12 h light/dark cycle at a controlled temperature of 23 ± 2 °C and 50 ± 5% humidity. All animals were acclimatized for 7 days with standard maintenance diet and water ad libitum. After acclimation, Kunming mice were randomly divided into four groups and treated with single intragastric administration: the Blank group (0.9% saline solution, 200 μL), the CS group (F-CS, 30 mg/kg BW, 200 μL), the CS-Fe group (F-CS-Fe, 30 mg/kg BW, 200 μL), and the LCS-Fe group (F-LCS-Fe, 30 mg/kg BW, 200 μL). Each treatment group included 24 mice, and four replicates (mice) were collected at each predetermined sacrifice time point (*n* = 4). Mice were sacrificed at 0.5 h, 1 h, 2 h, 4 h, 6 h, and 12 h after oral administration, then blood (1 mL) and individual intact tissues (heart, lung, liver, stomach, spleen, kidney, and intestine) were collected and subsequently processed under light-protected conditions. The blood was immediately centrifuged for 10 min at 4000 rpm, the plasma concentrations of F-CS, F-CS-Fe, and F-LCS-Fe were determined using a fluorescence microplate reader (Infinite F200 PRO, Tecan, Männedorf, Switzerland). The standard curves were prepared using different concentrations of F-CS, F-CS-Fe, and F-LCS-Fe. The intact tissues were fluorescently imaged using an imaging system (MIIS XFP-BIX, Molecular Devices, San Jose, CA, USA). Additionally, ImageJ software (version number: 1.54g) was performed to detect the region of interest (ROI), which represented the fluorescence intensity of mouse tissues.

Pharmacokinetic parameters of F-CS, F-CS-Fe, and F-LCS-Fe were calculated from the plasma concentration-time profiles, which were verified with a non-compartmental model using Phoenix WinNonlin software (version 8.1) as described previously [18]. The following parameters were determined, including the maximum plasma concentration (C_max_), the maximum plasma time (T_max_), the area under the blood concentration-time curve (AUC), and the half-time of elimination (T_1/2_)

### 2.8. Cell Culture and Viability Assays

RAW 264.7 cells were cultured in cell complete medium (minimum essential medium (MEM) complemented with fetal calf serum (20%, *v*/*v*) and penicillin/streptomycin (1%, *v*/*v*). RAW 264.7 cells were seeded into 96-well plates at a density of 0.5 × 10^4^ cells/well and incubated for 24 h. Then the supernatant was removed, and the cells were treated with 100 μL of CS, CS-Fe, and LCS-Fe at different concentrations (0–1000 μg/mL) for 24 h. Then the cells were treated with 20 μL of 3-(4,5-dimethylthiazol-2-yl)-2,5-diphenyltetrazolium bromide (MTT) solution (5 mg/mL in PBS) and incubated at 37 °C for 4 h. After discarding the supernatant, cells were treated with 150 μL of dimethyl sulfoxide (DMSO) for 10 min and subsequently subjected to measurement at a wavelength of 570 nm. The experiment was conducted with four replicates (*n* = 4). Cell viability was expressed as a percentage relative to the control group.$$Cell\;viability\;\left(\%\right)=\frac{{A_{s}-A}_{b}}{A_{c}-A_{b}}\times 100\%$$
where *A_s_* and *A_c_* were the absorbances of treated cells and untreated cells (control). Additionally, *A_b_* represented the absorbance of the blank group without cells.

### 2.9. Determination of Nitric Oxide (NO) Production

RAW 264.7 cell suspensions (1 × 10^5^ cells/mL) were seeded into the 96-well plates. After incubation for 24 h, the supernatant was removed, and the cells were treated with 1 µg/mL of LPS for 12 h. Then supernatant was removed, and the cells were treated with CS, CS-Fe, and LCS-Fe at concentrations of 200 μg/mL for 12 h. The NO concentrations in the supernatant of the experimental cells were determined by the NO assay kit (Beyotime Biotechnology Co., Ltd., Shanghai, China) referring to the manufacturer’s protocols. The experiment was conducted with four replicates (*n* = 4).

### 2.10. Determination of Inflammatory Factor Gene Expression by RT-PCR

RAW 264.7 cell suspensions (1 × 10^6^ cells/mL) were seeded into 6-well plates and incubated for 24 h. After incubation, the supernatant was removed and the cells were treated with 2 mL LPS (1 µg/mL) for 12 h. Then the supernatant was removed and the cells were treated with CS, CS-Fe, and LCS-Fe at concentrations of 200 μg/mL for 12 h. Total RNA was extracted from cells according to the defined methods of the SevenFast total RNA extraction kit (Saiwen Innovation Biotechnology Co., Ltd., Beijing, China). Subsequently, reverse transcription was performed using a Prime Script™ RT Master Mix (Takara Biomedical Technology Co., Ltd., Dalian, China). RT-PCR reactions were performed using TB Green^®^Premix Ex Taq™ II kit (Takara Biomedical Technology Co., Ltd., Dalian, China) and analyzed through a Real-Time PCR detector (Applied Biosystems™ QuantStudio™ 3&5, Thermo Fisher Scientific, Waltham, MA, USA). Inflammatory mediators such as nitric oxide synthase (iNOS) and cyclooxygenase-2 (COX-2), as well as the pro-inflammatory cytokines including IL-6, TNF-α, and IL-1β, were examined for their gene expression levels. The experiment was conducted with four replicates (*n* = 4). For primer sequences used, data refer to Table 1.

### 2.11. Targeted Metabolite Analysis

The metabolite profiles in LPS-activated RAW 264.7 cells were determined by the method reported with minor modifications [19]. The RAW 264.7 cells were cultured and treated as Section 2.10. After washing three times with PBS (4 °C), the cells were lysed and broken with liquid nitrogen. Then, cell metabolites were extracted using 80% methanol containing undecanoic acid, nonadecanoic acid, and L-glutamic acid-*d*_3_. Subsequently, the extract was vortexed and centrifuged at 12,000 rpm for 15 min, and the supernatant was filtered through a 0.22 μm organic membrane. The metabolites were analyzed using an HPLC system (Shimadzu, Kyoto, Japan) coupled with an AB Sciex triple 4000 QTRAP (AB Sciex, Framingham, MA, USA). Separation was performed with a BHE Amida column (150 × 2.1 mm, 2.5 μm), and the mobile phase A and mobile phase B were 20 mM ammonium acetate aqueous solution-acetonitrile (83:17, *v*/*v*; pH 9) and acetonitrile, respectively. The injection volume, flow rate, and column temperature were 1 μL, 0.4 mL/min, and 40 °C, respectively. Appendix A presented the mass spectrometry parameters for targeted metabolomics analysis, including precursor ions (Q1), product ions (Q3), dwell time, and collision energy (CE). Appendix A presented absolute quantification of each analyzed metabolite. The quantification of chromatographic peaks required a minimum peak height of 1000 cps and a peak area exceeding 500 cps*sec. The experiment was conducted with six replicates (*n* = 6).

### 2.12. Statistical Analysis

SPSS 24.0 statistical software was used for statistical analysis. The results were expressed as mean ± standard error of the mean. One-way ANOVA and Student’s *t*-test were used for comparison between groups. Differences between the experimental groups were considered statistically significant if *p* < 0.05.

## 3. Results and Discussion

### 3.1. Characterization of CS, CS-Fe, LCS-Fe, and Their Fluorescently Labeled Derivatives

Fluorescein labeling enables the visualization of polysaccharides, facilitating real-time tracking of their dynamic behavior during the absorption process. The amino group of 5-AF can react with the carboxyl groups in biomolecules and is commonly used for fluorescence labeling and analytical research of glycosaminoglycans [20]. In the present study, 5-AF was applied to label CS, CS-Fe, and LCS-Fe, followed by characterization using fluorescence spectroscopy and high performance gel permeation chromatography (HPGPC). As displayed in fluorescence spectra (Figure 1A), the maximum emission wavelength (λ_em_) of 5-AF was 522 nm, while the emission wavelength maxima of F-CS, F-CS-Fe, and F-LCS-Fe were all 534 nm (Figure 1B–D). The redshift of maximum emission spectra suggested that 5-AF was successfully attached to the polysaccharides through covalent conjugation [21,22,23].

Moreover, the molecular sizes of CS, CS-Fe, LCS-Fe, and their 5-AF labeling derivatives were characterized using HPGPC with a refractive index detector (Figure 1E–G). As shown in HPGPC analysis, chromatogram results revealed that CS, CS-Fe, and LCS-Fe displayed a single peak with retention times of 21.61 min, 25.22 min, and 25.78 min, respectively, which were similar to the retention times (21.95, 25.22, and 25.00 min) of F-CS, F-CS-Fe, and F-LCS-Fe, respectively. HPGPC separates molecules according to the molecular size, and the retention time (R_t_) of chromatographic peaks is inversely proportional to the molecular size [24,25]. Notably, fluorescence labeling hardly had effect on the sizes of CS, CS-Fe, or LCS-Fe, due to limited fluorescence substitution degrees, which were 0.26%, 0.33%, and 0.36% in F-CS, F-CS-Fe, and F-LCS-Fe, respectively. The differences in retention times between CS, CS-Fe, and LCS-Fe were attributed to the variation in molecular size. Iron ion complexation caused CS-Fe to exhibit a contracted molecular structure and a reduced molecular size. According to our previous studies, LCS-Fe exhibited a reduced Mw and a compact molecular conformation [16]. In summary, depolymerization and iron complexation affected the molecular conformation and apparent molecular size of CS, which may influence the pharmacokinetic behavior of CS, CS-Fe, and LCS-Fe in vivo.

### 3.2. In Vivo Absorption of Fluorescently Labeled CS, CS-Fe, and LCS-Fe

Following oral administration, the food and drug components are absorbed into the bloodstream and distributed to the target organs, thereby exerting their therapeutic effects [26]. To elucidate the absorption performance of fluorescently labeled CS, CS-Fe, and LCS-Fe in vivo, the plasma concentrations–time profiles (Figure 2) were determined. The plasma levels of F-CS, F-CS-Fe, and F-LCS-Fe were obvious at 0.5 h after single intragastric administration, and reached maximum plasma concentrations (C_max_) at 1 h with the concentration values of 135.27 ± 236.82 μg/mL, 376.60 ± 214.10 μg/mL, and 415.16 ± 109.50 μg/mL, respectively. Interestingly enough, F-LCS-Fe presented higher plasma levels than that of F-CS-Fe and F-CS within 12 h post-administration, and manifested the most significant difference at 6 h (*p* < 0.01). Evaluation of plasma concentrations demonstrated superior absorption efficiency in F-LCS-Fe compared to F-CS-Fe, while superior absorption efficiency in F-CS-Fe compared to F-CS. The reason for the higher plasma levels of labeled LCS-Fe could be ascribed to the low Mw and small molecular size, thereby facilitating efficient permeation through biological barriers and subsequent absorption. On the other hand, although CS-Fe had not undergone degradation, the reduction in molecular size caused by iron complexation still enabled labeled CS-Fe to exhibit higher blood absorption efficiency than CS. Furthermore, F-LCS-Fe had a higher area under the plasma concentration-time curve (2449.8 μg/mL·h) and moderate elimination half-time (3.3 h) than that of CS (501.8 μg/mL·h, 2.1 h), suggesting great absorption and slow clearance in vivo after intragastric administration [27]. The bioavailability of high-molecular-weight polysaccharides is constrained by low intestinal permeability, necessitating microbiota-driven depolymerization and fermentation for trace absorption and utilization [28,29]. Similarly, studies reported that a small quantity of high-molecular-weight polysaccharides can be absorbed into the bloodstream, and modifications such as reducing Mw, altering hydrophobic properties, and utilizing absorption enhancers facilitate their enhanced systemic absorption [30,31]. In addition, the elevated concentrations of CS-Fe and LCS-Fe in blood circulation may pose a potential risk of iron-induced systemic toxicity. However, studies demonstrated that the complexes formed by *Angelica sinensis* polysaccharides, *Flammulina velutipes* polysaccharides, and inulin with iron ions exhibited no toxic effects in vivo [14,32,33]. Moreover, Remya et al. [34] and Kim et al. [35] demonstrated that surface modification via coating with biocompatible materials (dextran and dextrin) enhanced iron bioavailability while preventing iron toxicity, including cytotoxicity, genotoxicity, and immunotoxicity. Therefore, cautious engineering design and rigorous safety assessment are critical strategies to ensure the efficacy and safety of food fortification and oral absorption.

### 3.3. In Vivo Distribution of Fluorescently Labeled CS, CS-Fe, and LCS-Fe

To investigate the fate of labeled CS, CS-Fe, and LCS-Fe in a living system, fluorescence imaging technology was applied to visualize their tissue distribution in vivo. As illustrated in Figure 3A, mice without treatment showed extremely tiny autofluorescence, which was induced by the biological components such as nicotinamide adenine dinucleotide, flavins, collagen, elastin, lipofuscin, tryptophan, and melanin in organisms [36]. The tissues from the F-CS, F-CS-Fe, and F-LCS-Fe groups were collected and imaged at 0.5 h, 1 h, 2 h, 4 h, 6 h, and 12 h after intragastric administration. Notably, strong fluorescence signals were observed in the stomach, liver, and intestine, while little fluorescence was exhibited in the heart, lung, spleen, and kidney at 0.5 h post-administration. Then F-CS, F-CS-Fe, and F-LCS-Fe gradually accumulated in the liver and intestine, with the intensity of fluorescent signals increasing. Specifically, the fluorescence signals observed in the intestine gradually accumulated and peaked at 6 h, 4 h, and 2 h (red arrow), respectively, after administration of F-CS, F-CS-Fe, and F-LCS-Fe, and then gradually declined. ROI quantitative analysis of fluorescence signals distribution across each tissue was performed and presented in Figure 3B–H. The fluorescence intensities in the hearts and lungs of the administration groups had no difference with those of the blank group, suggesting that F-CS, F-CS-Fe, and F-LCS-Fe hardly reached these organs. Of note, the highest mean fluorescence intensity was observed in the stomach, followed by the intestine, liver, kidney, and spleen in the treatment group. The results suggested that F-CS, F-CS-Fe, and F-LCS-Fe could pass through the stomach and intestine after oral administration, then be accumulated in liver and spleen, and eventually be excreted by kidney. Compared with F-CS-Fe and F-LCS-Fe, F-CS remained the highest fluorescence intensity in the stomach and intestine, likely attributed to its longer residence time in the digestive tract and lower permeability in the mucous cell layer as a macromolecule [30]. Fu et al. [37] reported that Fe^3+^ typically induces conformational changes in polysaccharides, leading to the formation of a structure similar to ferritin. This compact spherical molecular conformation reduces molecular heterogeneity and size, facilitating intestinal mucus penetration and enhancing binding to polysaccharide-specific receptors on intestinal cells [10,38]. These results were consistent with the low plasma concentration of F-CS observed in Section 3.2. The mean fluorescence intensities of F-CS-Fe and F-LCS-Fe in liver were significantly higher than that of F-CS, which might be attributed to the liver being the primary storage organ for iron, resulting in the accumulation of iron-carrying complexes [39].

### 3.4. Anti-Inflammatory Activity of CS, CS-Fe, and LCS-Fe in RAW 264.7 Cells

Macrophages are widely distributed in various tissues throughout the body, such as the liver, spleen, intestines, and bones, where they play a pivotal role in immune responses. Using macrophages to simulate the in vitro inflammation is an effective strategy for exploring the anti-inflammatory behavior of potential therapeutic agents [40]. Therefore, the LPS-stimulated RAW 264.7 cell model was used to evaluate the anti-inflammatory effects of CS, CS-Fe, and LCS-Fe. Firstly, MTT assay was employed to detect the cell viability of RAW 264.7 cells (Figure 4A–C). Results showed that CS, CS-Fe, and LCS-Fe promoted the proliferation of macrophages at concentrations of 100 μg/mL-800 μg/mL, and the proliferation rates had a slight decrease at 1000 μg/mL for CS-Fe and LCS-Fe (*p* < 0.0001). Both cellular toxicity and abnormal proliferation can interfere with the normal physiological status of cells [41]. Therefore, concentrations of 100 μg/mL and 200 μg/mL were determined as safe doses that approximate the normal growth conditions of RAW 264.7 cells for subsequent anti-inflammatory activity experiments.

Upon encountering foreign antigens, the immune system of host triggers a cascade of responses, in which activated macrophages release a torrent of inflammatory mediators (NO, iNOS, and COX-2) and pro-inflammatory cytokines (IL-6, TNF-α, and IL-1β). As shown in Figure 4D, the NO production level was significantly increased in the LPS group (*p* < 0.0001), indicating that RAW 264.7 cells had already developed an inflammatory reaction due to the stimulation of LPS. Meanwhile, compared with the LPS group, CS only weakly reversed the inflammatory state, while CS-Fe and LCS-Fe significantly inhibited the secretion of NO. As displayed in Figure 4E,F, the iNOS and COX-2 gene expression exhibited a significant, dose-dependent decrease in RAW 264.7 cells after treatment with CS, CS-Fe, and LCS-Fe, which was consistent with the production trend of NO. Intervention with CS, CS-Fe, and LCS-Fe (200 μg/mL) significantly reduced the expression levels of iNOS by 58.8%, 74.9%, and 64.7%, and COX-2 by 20.5%, 23.2%, and 32.2%, respectively. NO was catalyzed and synthesized by a set of enzymes called inducible nitric oxide synthases (iNOS), and the high production of iNOS was often accompanied by up-regulated COX-2 in the inflammatory process [42].

Compared to the control group, the gene expression levels of IL-6, TNF-α, and IL-1β were significantly increased after LPS stimulation (Figure 4G–I). Notably, CS, CS-Fe, and LCS-Fe dose-dependently inhibited the gene expression levels of IL-6, while CS (100 μg /mL and 200 μg/mL) and LCS-Fe (200 μg/mL) exhibited stronger inhibitory effects. Similarly, the results of TNF-α and IL-1β indicated that CS-Fe and LCS-Fe significantly down-regulated the gene expression levels of pro-inflammatory cytokines in RAW 264.7 macrophages, with LCS-Fe at the concentration of 200 μg/mL showing the greatest reduction of 39.18% and 60.48%, respectively. Furthermore, studies have reported that the monosaccharide composition, molecular weight, and sulfate group content of polysaccharides were important factors for their anti-inflammatory effect [19,43]. Qiu et al. [43] and Kuang et al. [44] reported that oligosaccharides mediated the iNOS/NF-κB signaling axis to exert the highest anti-inflammatory activity. Wang et al. [2] reported that low Mw CS prepared by cleavage reaction had better anti-inflammatory activity than CS. These findings are consistent with our study demonstrating the enhanced anti-inflammatory activity of LCS-Fe. Furthermore, *Enteromorpha prolifera* polysaccharide-Fe and *Enteromorpha prolifera* polysaccharide-Zn played inflammation suppression roles by inhibition of the M1 macrophages [45]. The formation of metal complexes can change the structure of polysaccharides to improve biological properties.

### 3.5. Metabolism Regulation Effects of CS, CS-Fe, and LCS-Fe on LPS-Induced RAW 264.7 Cells

Assessment of the impact of CS, CS-Fe, and LCS-Fe on LPS-activated RAW 264.7 cells at the metabolite level. OPLS-DA analysis in positive and negative ion modes was utilized to delineate the distinct impacts among the Blank, Model, CS, CS-Fe, and LCS-Fe groups (Figure 5). The dense and concentrated sample distribution of each group along component Ⅱ direction indicated smaller differences within the same group [19]. The significant separation of the Blank group from the Model group along component I direction. Moreover, the metabolome of CS-Fe and LCS-Fe treatments was distinguished from the Model group along component I direction, illustrating a closer similarity to the Blank group. The regulatory effect of CS was not significant, which was consistent with the inhibition of NO and pro-inflammatory cytokines secretion.

A total of 42 significantly different metabolites were identified in the comparisons from Blank vs. Model, Model vs. CS, Model vs. CS-Fe, and Model vs. LCS-Fe, based on the criteria of VIP > 1.2 and *p* < 0.05. Subsequently, the clustering heatmap delineated the expression of 42 significantly different metabolites and its statistical significance (Figure 6). Comparing with the Blank group, the LPS treatment increased 26 metabolites, such as S-methyl-5-thioadenosine (MTA), proline, phenylalanine, valine, 3-hydroxybutyryl-CoA, 2-aminooctanoic acid, butyryl-CoA, homoserine, dephospho-CoA, hydroxyproline, leucine, isoleucine, homocysteine, methylcysteine, glycerophosphocholine, glycine, glutathione, glucosamine, guanosine triphosphate, citrulline, pipecolic acid, glutamate, deoxycytidine monophosphate (dCMP), creatine, alanine, and threonine, and decreased 11 metabolites in Model group, such as 2-methylnicotinamide, uridine monophosphate (UMP), guanosine monophosphate, acetylcarnitine, spermidine, 5-methoxytryptophan, betaine, carnitine, betaine aldehyde, NAD^+^, and NADH. Compared with the Model group, the LCS-Fe treatment significantly up-regulated 9 metabolites and down-regulated 8 metabolites, including the reversal of 10 metabolites of MTA, proline, valine, 2-aminooctanoic acid, isoleucine, pipecolic acid, glutamate, 2-methylnicotinamide, 5-methoxytryptophan, and betaine. In addition, compared with the Model group, the CS-Fe treatment significantly up-regulated 10 metabolites and down-regulated 9 metabolites, including the reversal of 12 metabolites of MTA, proline, valine, 2-aminooctanoic acid, hydroxyproline, leucine, isoleucine, 2-methylnicotinamide, guanosine monophosphate, betaine, NAD^+^, and NADH. Furthermore, the CS treatment significantly up-regulated 12 metabolites and down-regulated 8 metabolites, including reversal in 10 metabolites. Valine and leucine, as branched-chain amino acids, are known for activating P13K/PKB pathway and inducing Th2 immune responses and eosinophilic inflammation [46]. Enhanced proline synthesis occurs in macrophages under inflammatory conditions to maintain intracellular redox homeostasis and counteract hypoxia [47]. MTA is known to control the inflammatory microenvironment of tumor tissues and induce apoptosis [48]. Betaine exerts anti-inflammatory effects in LPS-induced RAW 264.7 cells through the inhibition of the NF-κB signaling pathway [49]. Therefore, CS, CS-Fe, and LCS-Fe reshaped the metabolic dysregulation induced by LPS, reversed the reprogramming-associated changes in metabolites involved in the inflammatory response, and supported the anti-inflammatory phenotype. In addition, the clustering branch distance indicated that the LCS-Fe group was most similar to the Blank group, and they both were markedly different from the Model group. On the contrary, the CS group showed the greatest difference compared to the Blank group.

To elucidate the significant alterations in the metabolism of LPS-stimulated RAW 264.7 cells, metabolic pathways were analyzed using KEGG database. As shown in Figure 7A, LPS treatment markedly influenced the biosynthesis of valine, leucine, and isoleucine, the metabolism of glycine, serine, and threonine, the metabolism of arginine and proline, and so on. However, CS, CS-Fe, and LCS-Fe treatments significantly restored the biosynthesis of valine, leucine, and isoleucine. Interestingly, LCS-Fe treatment also significantly regulated the metabolism of glutathione between Model group and LCS-Fe group. Consistent with the metabolite profiles in the heatmap, CS, CS-Fe, and LCS-Fe reduced the contents of valine and leucine in the RAW 264.7 cells by regulating the biosynthesis of valine, leucine, and isoleucine, thereby suppressing the inflammation (Figure 7C,D). Additionally, the glutathione metabolism pathway is closely related to the repair and antioxidant functions of cells [50,51,52]. In summary, the observed differences in metabolites and enriched metabolic pathways may be attributed to variations in their molecular weights [53]. The anti-inflammatory activity of LCS-Fe may be closely related to its efficient antioxidant capacity, which can maintain intracellular redox balance [16].

## 4. Conclusions

In the present study, chondroitin sulfate-iron complex (CS-Fe) and low-molecular-weight chondroitin sulfate-iron complex (LCS-Fe) were synthesized, and they both demonstrated significantly higher plasma concentration and liver accumulation in vivo compared to original CS following oral administration in mice, with LCS-Fe showing the highest C_max_ of 415.16 ± 109.50 μg/mL. Furthermore, CS, CS-Fe, and LCS-Fe all significantly inhibited the inflammatory factors, including NO, COX-2, and IL-1β, in macrophages induced by LPS, and CS-Fe and LCS-Fe showed even better effects. Furthermore, metabolomics analysis indicated that LPS stimulation of macrophages induced reprogramming-associated changes in 37 metabolites to support the pro-inflammatory phenotype. However, CS, CS-Fe, and LCS-Fe were found to reverse the levels of 10, 12, and 10 of these metabolites, respectively, making the clustering pattern of LCS-Fe most similar to that of the control group, followed by CS-Fe. KEGG enrichment analysis of differential metabolites indicated the biosynthesis of valine, leucine, and isoleucine as the most prominent pathway influenced by LPS treatment, and it was reversed by CS, CS-Fe, and LCS-Fe. These findings provide valuable insights for the application and development of CS products.

## Figures and Tables

**Figure 1 foods-14-03356-f001:**
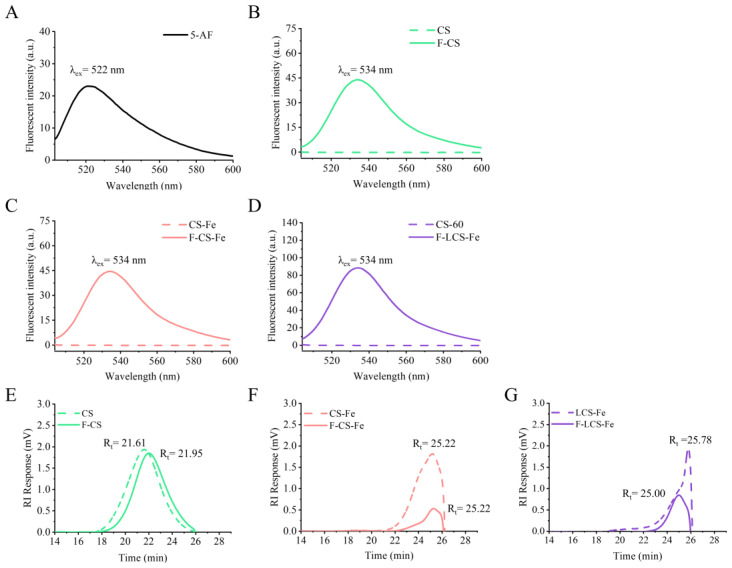
The fluorescence spectrums of (**A**) 5-AF, (**B**) CS and F-CS, (**C**) CS-Fe and F-CS-Fe, (**D**) LCS-Fe and F-LCS-Fe. HPGPC chromatograms of (**E**) CS and F-CS, (**F**) CS-Fe and F-CS-Fe, (**G**) LCS-Fe and F-LCS-Fe.

**Figure 2 foods-14-03356-f002:**
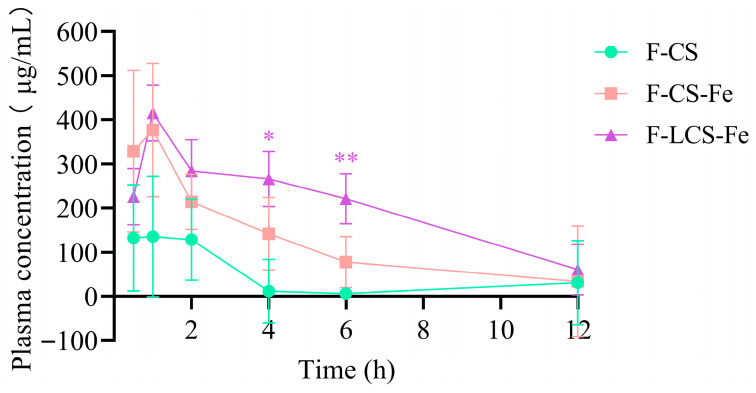
The plasma concentration–time profiles of F-CS, F-CS-Fe, and F-LCS-Fe after single intragastric administration in mice (each point represents mean ± SEM, *n* = 4). * *p <* 0.05 and ** *p <* 0.01 indicate statistically significant differences.

**Figure 3 foods-14-03356-f003:**
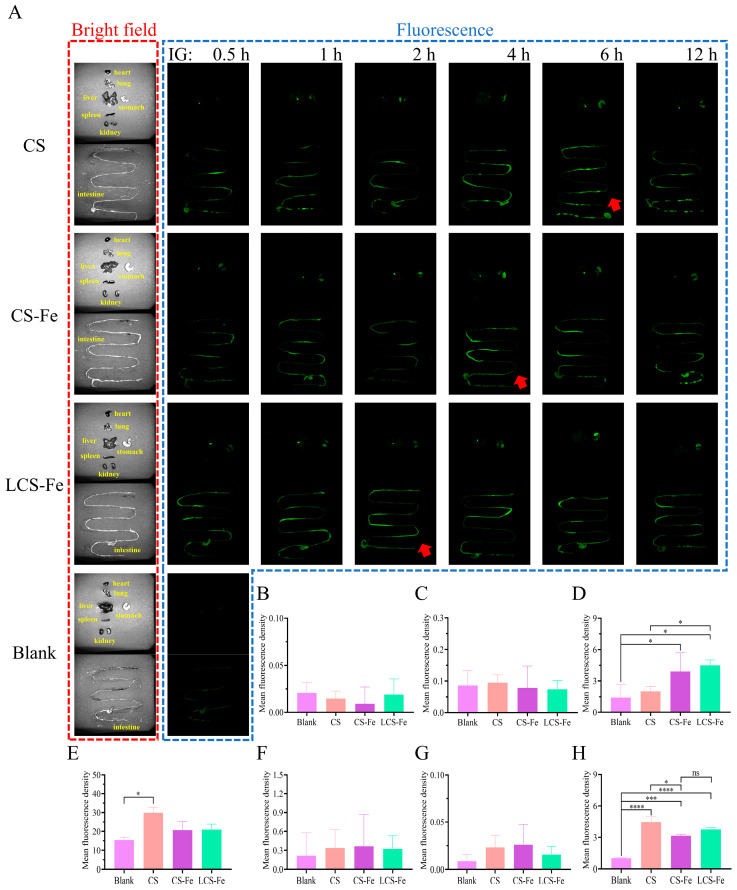
(**A**) In vivo distributions of F-CS, F-CS-Fe, and F-LCS-Fe after single intragastric administration. The black-and-white images in the first column represented different tissues captured using a bright field channel (organ names were labeled in yellow text). The fluorescence images were acquired using the 470Ex-525/50Em fluorescence channel, with the tissue location consistent with that observed in the bright field. ROI values in (**B**) heart, (**C**) lung, (**D**) liver, (**E**) stomach, (**F**) kidney, (**G**) spleen, and (**H**) intestine at 6 h after single intragastric administration (*n* = 4). * *p <* 0.05, *** *p <* 0.001, and **** *p <* 0.0001 indicate statistically significant differences. “ns” indicates not statistically significant differences.

**Figure 4 foods-14-03356-f004:**
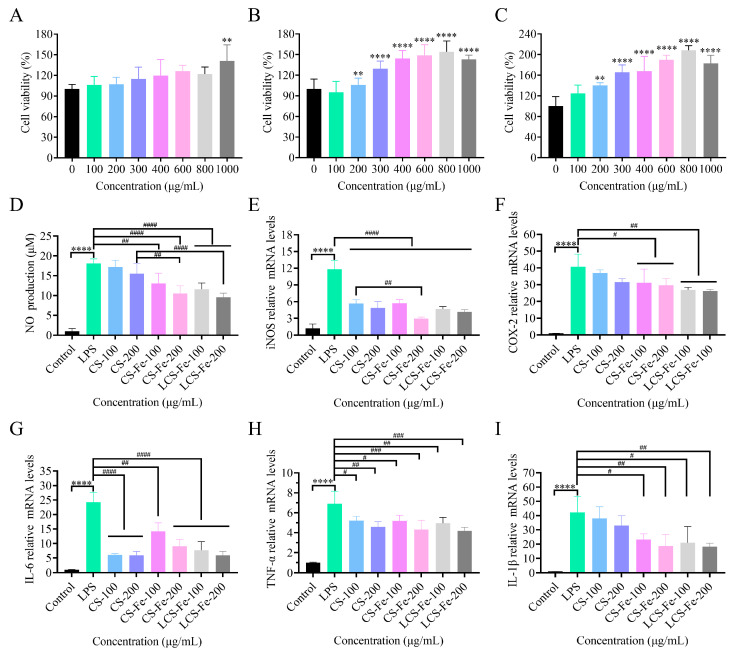
Cell viability under different concentrations of (**A**) CS, (**B**) CS-Fe, and (**C**) LCS-Fe. The effects of different concentrations of CS, CS-Fe, and LCS-Fe on the production of inflammatory mediators and pro-inflammatory cytokines in LPS-induced RAW 264.7 cells. (**D**) NO production; (**E**) iNOS mRNA expression levels; (**F**) COX-2 mRNA expression levels; (**G**) IL-6 mRNA expression levels; (**H**) TNF-α mRNA expression levels; (**I**) IL-1β mRNA expression levels. Statistically significant differences were indicated with ^#^ *p <* 0.05, ** *p* < 0.01, ^##^ *p <* 0.01, ^###^ *p <* 0.001, **** *p* < 0.0001 and ^####^ *p <* 0.0001.

**Figure 5 foods-14-03356-f005:**
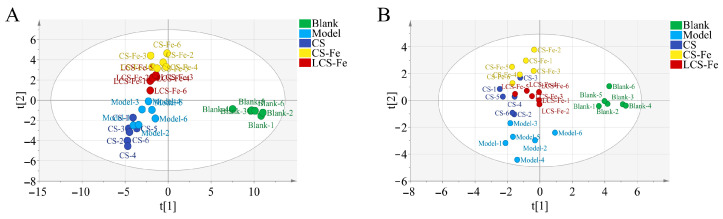
OPLS-DA analysis of metabolite profiles among different groups in (**A**) positive and (**B**) negative models.

**Figure 6 foods-14-03356-f006:**
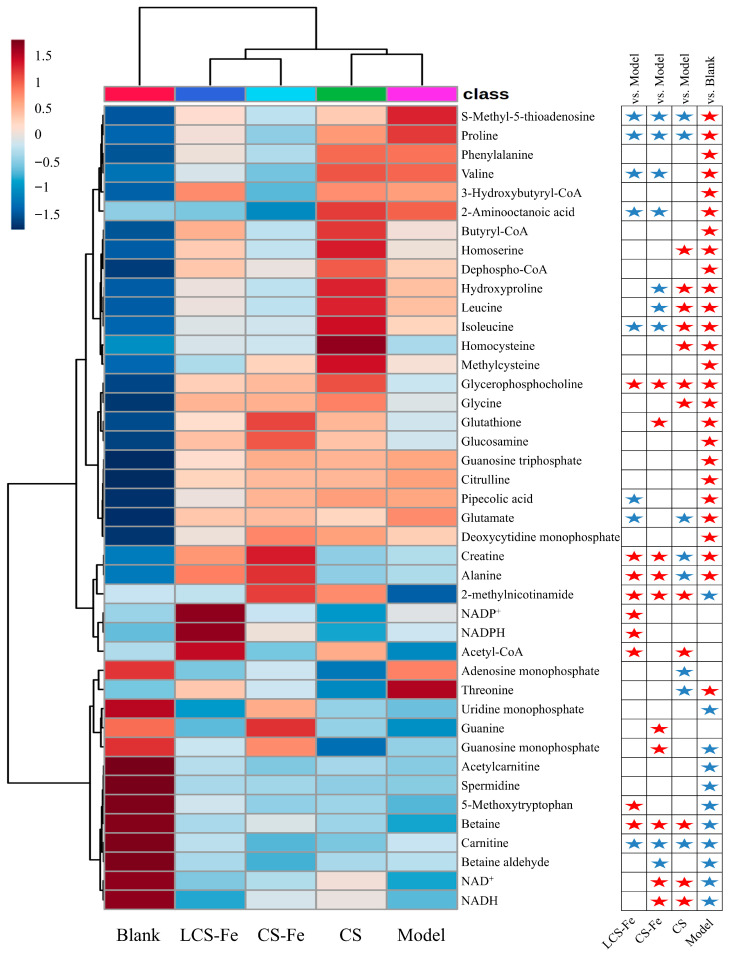
Clustering heatmap of the differential metabolites among different groups. Red stars represent significantly up-regulated metabolites (Model vs. Blank; CS vs. Model; CS-Fe vs. Model; LCS-Fe vs. Model; VIP ≥ 1.2 & *p* < 0.05), and blue stars represent significantly down-regulated metabolites (Model vs. Blank; CS vs. Model; CS-Fe vs. Model; LCS-Fe vs. Model; VIP ≥ 1.2 & *p* < 0.05).

**Figure 7 foods-14-03356-f007:**
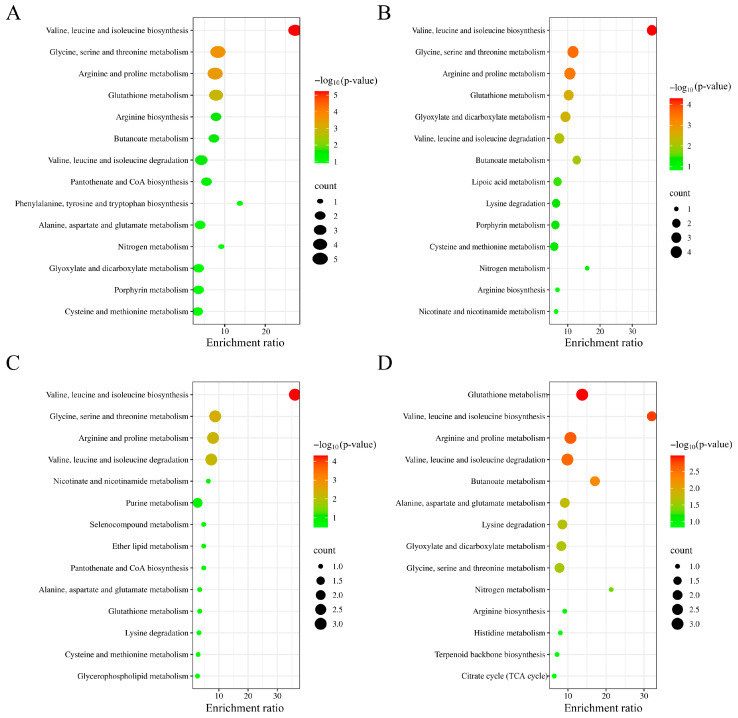
KEGG enrichment scatter plot of (**A**) Blank group vs. Model group, (**B**) Model group vs. CS group, (**C**) Model group vs. CS-Fe group, (**D**) Model group vs. LCS-Fe group.

**Table 1 foods-14-03356-t001:** Primer sequences and their fragment information.

Gene	Forward Primer (5′→3′)	Reverse Primer (5′→3′)
iNOS	ATGTCCGAAGCAAACATCAC	TAATGTCCAGGAAGTAGGTG
COX-2	CAGCAAATCCTTGCTGTTCC	TGGGCAAAGAATGCAAACATC
TNF-α	GATCGGTCCCCAAAGGGATG	GGCTACAGGCTTGTCACTCG
IL-1β	TTCATCTTTGAAGAAGAGCCCAT	TCGGAGCCTGTAGTGCAGTT
IL-6	TGGAAATGAGAAAAGAGTTGTGC	CCAGTTTGGTAGCATCCATCA

## Data Availability

The original contributions presented in this study are included in the article. Further inquiries can be directed to the corresponding author.

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
