# Peer review of "In Vivo Absorption of Iron Complexes of Chondroitin Sulfates with Different Molecular Weights and Their Anti-Inflammation and Metabolism Regulation Effects on LPS-Induced Macrophages"

_foods, 2025, doi:10.3390/foods14193356_

Round 1

Reviewer 1 Report

Comments and Suggestions for Authors

The manuscript entitled: “In Vivo Absorption of Iron Complexes of Chondroitin Sulfates with Different Molecular Weights and Their Anti-Inflammation and Metabolism Regulation Effects on LPS-Induced Macrophages”. It presents an interesting study on the application of metal complexes in the anti-inflammatory response. The paper is very complete, with minor comments to improve the quality of the article.

145-147. Indicate how many animals were assigned per group/time, indicate the volume of administration, specify the volume of blood obtained from each animal. Was the blood processed immediately? The same applies to tissues; include a description and preservation of the tissues.

Figure 3. Control: in the image of the organs, the lung is not named. Include in the figure caption the n that each bar represents.

Figure 4A-C. Correct the title of the "y" axes.

Reviewer 2 Report

Comments and Suggestions for Authors

The paper addresses the bioavailability of chondroitin sulfates and their role in regulating LPS-induced macrophage metabolism. The topic is highly relevant; however, providing additional details and clarifications could further enhance the quality and impact of the manuscript.

Including or citing relevant in vivo toxicity tests could help complement the available data and strengthen the manuscript.

Why did the authors choose to analyze gene expression levels rather than performing protein analysis?

The following points are presented by topic for the authors’ consideration:

Item 2.8; 2.9; 2.10 - The authors should clarify how many replicates were performed for the macrophage experiments.

Line 186 – Was the SevenFast Total Protein Extraction Kit used, or was an RNA extraction kit employed instead?

Item 2.7 – How many animals were used in each group and at each time point following oral administration? The authors should clarify how the tissues were prepared prior to analysis. Were the tissues sectioned (e.g., sliced) prior to analysis, and if so, how was this procedure performed?

Item 2.11 - The authors are encouraged to specify and cite the targeted metabolites in the methodology to improve clarity and reproducibility.

Item 3.1 – Was the use of standard molecular weights considered to estimate the weights of CS, F-CS, and F-LCS?

Figure 3 – The legends are not clear. Could the authors clarify the information provided and improve the figure quality?

Figure 4  and  line 323 – Do panels A, B, and C refer to “cell viability” or “cell death”? As all results appear to be above 80%, the rationale for selecting low concentrations to proceed should be clarified.

Line 383 - For clarity and validation, chromatogram/spectrum of the 42 metabolites analyzed should be provided in the manuscript for each treatment group. For completeness, the manuscript should report the quantification limits and include absolute quantification of each analyzed component.

Reviewer 3 Report

Comments and Suggestions for Authors

Dear Authors,

First of all, I would like to congratulate you on the excellent manuscript entitled “In Vivo Absorption of Iron Complexes of Chondroitin Sulfates with Different Molecular Weights and Their Anti-Inflammation and Metabolism Regulation Effects on LPS-Induced Macrophages”, submitted to Foods (MDPI).

Below, I provide some comments and suggestions that may contribute to enhancing the clarity and consistency of the manuscript:

  • Line 40: It is recommended to revise the wording, as the current text may lead to misinterpretations. Chondroitin 4-sulfate (CS-A) exhibits similar levels of sulfation at the 4 and 6 positions of N-acetyl-galactosamine, whereas chondroitin 6-sulfate (CS-C) predominantly (≈90%) contains sulfation at the 6 position of the same unit.

  • Materials and Methods: The type of chondroitin sulfate used in the experiments is not clearly specified. It would be relevant to provide details regarding its origin, purity, and characterization, thereby reinforcing the reproducibility of the study.

  • Line 152: Please clarify the meaning of the value “154 g,” as, in its current form, it generates ambiguity.

  • Molecular weight and complexation: As mentioned in the manuscript, the formation of the CS-Fe complex leads to structural modifications, resulting in molecular contraction and, consequently, an apparent reduction in molecular weight. However, it is essential to note that, assuming no depolymerization of CS has occurred, what is observed is a decrease in the spatial volume occupied by the molecule, rather than in its actual molecular weight. In this context, it should be emphasized that direct comparisons between CS-Fe and LCS-Fe (globular structures) and linear CS may not be appropriate. Additionally, was the quantification of uronic acid or hexosamine performed for these compounds to provide further parameters of characterization?

  • Intestinal diffusion: Considering that globular structures may display greater intestinal diffusivity when compared with linear CS, a brief discussion on this aspect would be valuable.

  • Anti-inflammatory activity: The literature reports that the anti-inflammatory activity of CS is associated with the inhibition of NF-κB activation, with a consequent reduction in its nuclear translocation. It would be important to clarify whether this effect was also observed for the iron-complexed compounds in the present study.

In summary, this is a high-quality work that presents original and scientifically relevant findings. The above suggestions aim to enhance the clarity and consistency of the manuscript.

Sincerely
